# Thyroid hormone deficiency during zebrafish development impairs central nervous system myelination

**Brenda Minerva Farías-Serratos[1], Iván Lazcano[1], Patricia Villalobos[1], Veerle M. Darras[1,2], Aurea Orozco[1]***

1 Instituto de Neurobiología, Universidad Nacional Autónoma de México (UNAM), Querétaro, Qro., México,
2 Biology Department, Laboratory of Comparative Endocrinology, KU Leuven, Leuven, Belgium

* aureao@unam.mx

**Data Availability Statement:** All relevant data are within the manuscript and its Supporting Information files.

## Abstract

Thyroid hormones are messengers that bind to specific nuclear receptors and regulate a wide range of physiological processes in the early stages of vertebrate embryonic development, including neurodevelopment and myelogenesis. We here tested the effects of reduced T3 availability upon the myelination process by treating zebrafish embryos with low concentrations of iopanoic acid (IOP) to block T4 to T3 conversion. Black Gold II staining showed that T3 deficiency reduced the myelin density in the forebrain, midbrain, hindbrain and the spinal cord at 3 and 7 dpf. These observations were confirmed in 3 dpf *mbp*:*egfp* transgenic zebrafish, showing that the administration of IOP reduced the fluorescent signal in the brain. T3 rescue treatment restored brain myelination and reversed the changes in myelin-related gene expression induced by IOP exposure. NG2 immunostaining revealed that T3 deficiency reduced the amount of oligodendrocyte precursor cells in 3 dpf IOP-treated larvae. Altogether, the present results show that inhibition of T4 to T3 conversion results in hypomyelination, suggesting that THs are part of the key signaling molecules that control the timing of oligodendrocyte differentiation and myelin synthesis from very early stages of brain development.

## Introduction

Thyroid hormones (THs) are essential regulators of growth and development in various tissues, including the brain [1,2]. Most of these processes occur through binding to specific nuclear receptors known as TH receptors or TRs. They function as hormone-activated transcription factors that control the expression of a large number of target genes. For these genomic actions, thyroxine or T4 acts primarily as a prohormone while T3 is the major ligand that binds to different TR isoforms [2,3]. During neurodevelopment, THs participate in neuronal proliferation, migration, differentiation and signaling, as well as in brain myelination [4]. Indeed, TH deficiency during the mammalian fetal or early postnatal periods may cause irreversible mental retardation and neurological deficits [5,6]. Furthermore, the hypothyroid

**Funding:** BMFS, CONACyT fellowship 573131 AO, PAPIIT, UNAM IN204920 VMD, Research Foundation – Flanders (K203420N) VMD, Programa de Estancias de Investigación (PREI) DGAPA, UNAM The funders had no role in study design, data collection and analysis, decision to publish, or preparation of the manuscript.

**Competing interests:** The authors have declared that no competing interests exist.

brain presents delayed myelination and poor deposition of myelin, whereas hyperthyroidism during the early postnatal period results in accelerated myelination [7–10].

Myelin is an axonal insulator that facilitates the propagation of action potentials and myelination is the last major event to occur during central and peripheral nervous system development of all vertebrates except agnathans [11–13]. In mammals, myelination begins in late stages of embryonic development and ends in the postnatal stages [4] while in the zebrafish (*Danio rerio*) it starts around 2.5 days post fertilization (dpf) [14–16]. This is why the zebrafish has become a popular model organism to uncover novel mechanisms controlling vertebrate myelination [17,18] and therefore also its TH dependency.

The fundamental structure of myelin is conserved in vertebrates. Homologs for myelin proteins such as myelin basic protein (MBP), myelin protein zero (MPZ), and proteolipid protein (PLP1) are present in zebrafish. It is currently accepted that these three structural proteins are distributed in both the central- (CNS) and the peripheral nervous systems [17,19]. In the CNS, neural precursor cells give rise to oligodendrocyte precursor cells (OPCs), which proliferate as they migrate through the developing CNS. Following migration, OPCs transition into pre-myelinating oligodendrocytes (OLs) which upon terminal differentiation become myelinating OLs and iteratively wrap their plasma membranes around multiple axon segments to form myelin sheaths [16]. Nascent myelin sheaths are evident around 2.5 dpf in the zebrafish CNS [18,20]. By 3 dpf, myelin is detected in the ventral hindbrain and robustly myelinated bundles of axons are observed by 7 dpf [17,21], extending rostrally over the following days to the cerebellar plate, midbrain, and optic nerve [22]. The transcription factors *olig1/2* control OL linage progression in early stages and subsequently, OL differentiation [14]; *olig2* transcripts are present in the embryo [23] and protein expression starts around 9 hours post-fertilization (hpf) in the ventral region of the neural plate [12]. The transcription factor *sox10* is specifically required for survival of cells committed to myelination and its expression starts around 10.5 hpf in neural precursors [16]. Among zebrafish myelin-related genes, at least *olig2*, *mbp* and *mpz* have been shown to be direct T3-responsive genes [24], like their mammalian counterparts [25–27]. Thus, it is possible that the stimulatory effects of THs on myelination first lead to OL maturation and then to the synthesis of myelin structural proteins by myelinating OLs; both events would involve genomic paths mediated by TRs.

During the first days post-fertilization in teleost embryos, including zebrafish do not synthetize TH but the fertilized eggs contain a load of T4, which is gradually converted to T3 by the activity of deiodinases [28]. Here, we blocked conversion of maternal and later of larval T4 to T3 through treatment with IOP, a drug proven to inhibit deiodinase activity in mammals as well as in non-mammalian vertebrates, including teleosts [29–31] to analyze the effects of the induced T3 deficiency on myelination. Using myelin- and OPC staining techniques as well as transgenic *mbp*:*egfp* zebrafish larvae, we here show that T3 deficiency reduced the amount of myelin in the analyzed brain areas, possibly by reducing the amount of OPCs, as suggested by a lower NG2 immunoreactivity.

## Materials and methods

### Zebrafish

Wild-type zebrafish embryos obtained from crosses of adult male and female zebrafish were used, collected at the one cell stage and kept under standard conditions at 28°C in E3 embryonic medium (5 mM NaCl, 0.17 mM KCl, 0.33 mM CaCl2 and 0.33 mM MgSO4 and methylene blue), with 14:10 h light:dark cycles. After 4 dpf, larvae were fed twice a day with an interval of at least 6 hours between meals. During each feeding, the larvae were in contact with the food flake for 1 hour, and then the E3 medium was replaced. This study was carried out in

strict accordance with the national and international guidelines for the care and use of laboratory animals. The protocol was approved by the Ethics for Research Committee of the Instituto de Neurobiología of the Universidad Nacional Autónoma de México (Protocol Number: 095A). Euthanasia was performed through hypothermal shock by immersion in ice water (2–4˚C), and all efforts were made to minimize suffering.

## Quantitative PCR

RNA was extracted in control and treated groups with Trizol Reagent (Life technologies). Retrotranscription was performed with M-MLV Reverse Transcriptase (Promega) from 500 ng of total RNA and 0.5 μg oligo (dT). The used oligonucleotides (**S1 Table**) were as previously published [23] and designed with real time PCR tool from IDT (Integrated DNA Technologies). Primer specificity was confirmed by agarose gel electrophoresis and melting curve analysis. In all cases, reactions contained 1 μL of reverse transcribed product, Maxima SYBR Green/ROX qPCR Master Mix (Thermo Scientific), and oligonucleotides at a final concentration of 0.5 μM. A Step One instrument was used for detection and data analysis according to the manufacturer's instructions (Applied Biosystems). For all genes, amplicons obtained by RT-PCR were cloned (pJET1.2/blunt Cloning Vector; Thermo Scientific) and sequenced. These constructs were used to prepare standard curves ranging from $10^1$ to $10^9$ molecules/μl. Absolute quantifications were obtained interpolating the Ct value of each experimental sample in the standard curve of the corresponding gene and normalizing with reference genes β-actin and LSM. The absolute mRNA concentration was expressed as molecules per microgram of total mRNA used in the RT reaction. Identical PCRs from the RNA samples before the reverse transcription reaction yielded no detectable products.

## Alcian blue staining

Alcian blue staining was performed in zebrafish larvae as previously reported [32]. Briefly, 7 dpf larvae were anesthetized, immersed overnight in PFA 4% and washed in PBS 0.1% Tween-20. Animals were bleached in 30% hydrogen peroxide until eye depigmentation was evident. Again, embryos were washed in PBS-Tween and transferred into 1% of hydrochloric acid, 70% ethanol, and 0.1% Alcian blue overnight. Finally, samples were cleared in acidic ethanol for 4 hours, dehydrated in ethanol series and stored in glycerol. Pictures were captured with a Leica DM500 microscope at 10X and LAS EZ 3.0 software.

## Myelin staining with Black-Gold II

Fixation of zebrafish larvae was performed immediately after euthanasia by immersion in freshly prepared 4% paraformaldehyde (PFA) and then incubated overnight at 4˚C. The next day the larvae were washed 5 times with 1x PBS for 5 min at room temperature (RT). Subsequently, they were cryoprotected with 10, 20 and 30% sucrose solutions in water and incubated overnight again. Whole-mount Black-Gold II staining was performed using a modified protocol [33], in which the larvae were briefly rehydrated with 1x PBS for 2 min at RT; then incubated in 0.3% Black-Gold II solution at 60˚C until staining was complete and afterwards were washed 5 times with 1x PBS for 2 min at 60˚C. Finally, larvae were dehydrated with a graduated series of alcohol solution, rinsed with xylene, and stored at -20˚C in Eppendorf tubes.

For image analysis, each zebrafish larva was embedded in 1.5% low melting point agarose, examined in sagittal position and Rostral-Caudal (R-C) orientation with the LEICA microscope objectives of 4x and 40x and with the integrated ICC50 HD color camera. Zebrafish brain 40x images were analyzed with Image-Pro Plus 6 (IPP6) as follows: myelin cords stained in violet color with Black-Gold II [34] were selected and the density of myelin fibers was

quantified based on the number of positive pixels in the selected area. IPP6 allows to select a specific color (violet) in the tinted image, but without considering their intensity. All regions were selected manually.

### OPC staining

Immunostaining was performed according to [35] using primary antibodies shown to recognize fish NG2 [36]. Briefly, zebrafish larvae were fixed overnight with PFA 4%, dehydrated in methanol series and stored at -20˚C. For immunostaining, larvae were rehydrated, washed with PBS and incubated in a 3% $H_2O_2$/0.5% KOH medium for 40 min to decrease pigmentation. To unmask antigens, larvae were incubated in sodium-citrate (pH 6.0) for 30 min. Additionally, larvae were submerged in acetone 100% for 30 min, washed with sterile water for 5 min and incubated with a blocking solution (PBS triton 1% plus BSA 0.1%) for 30 min. The primary antibody anti-NG2 (1:100, Millipore, USA, Cat. no. AB5320) was incubated for two days at 4˚C. The secondary antibody (goat anti-rabbit IgG, 1:250, Invitrogen, USA, Cat. No. A11008) was incubated overnight at 4˚C.

For image capture, zebrafish larvae were oriented dorsally onto 1.5% low melting point agarose in concave slides and the top of the head was microphotographed. Fluorescence was detected with a Carl Zeiss-LSM 700 confocal microscope and ZEN software. The images were obtained at 10x and were scanned three-dimensionally at a 114 μm depth in set Z-scan (0.66 pixels) at a resolution of 512 × 512 pixels. The Image J program from the Fiji package was used for data processing.

### *mbp*:*egfp* transgenic fish

To generate embryos that express *egfp* under the *mbp* promoter, we used the Tol2 transposon system [37]. To this end, transposase mRNA was synthetized *in vitro* using a Not1-linearized pCS-zT2TP plasmid (gift from Dr. K. Kawakami) as template in an mMESSAGE mMACHINE SP6 kit (Life Technologies). Transposase mRNA was purified by ethanol precipitation and resuspended in RNase-free water. One cell stage F0 zebrafish embryos were injected [38] no later than 20 minutes after fertilization, with a volume of 1 nL of working solution containing 25 pg of transposase mRNA and 25 pg of plasmid Tg (*mbp*:*egfp*) [39]. These embryos were collected and submitted to IOP treatment (0.5, 1, 1.5 and 2 μM) for 3 dpf. The same approach has been used previously to visualize GFP expression in F0 zebrafish (e.g. [40,41]). To take into account the variability due to the inherent mosaicism in F0 transgenic fish, we analyzed at least 15 animals per group allowing to show clear effects of the treatments.

After fixation (see above), fluorescence was detected as described for OPC staining with the following changes: The images were obtained at 20x and zebrafish heads were scanned three-dimensionally at a 260 to 300 μm depth in set Z-scan (0.66 pixels) at a resolution of 512 × 512 pixels. The area of interest was manually segmented, and the intensity threshold was matched for all the scanned images. The heads of the larvae were reconstructed using the Amira-Avizo software for 3D visualization and the values were reported in voxels.

### IOP treatment and T3 rescue

We first tested different doses of IOP to determine the lowest effective concentration (**S1 Fig**). For further experiments, wild-type larvae were subjected to one of the following 4 treatments: solvent control, IOP 0.5 μM, T3 10 nM and IOP 0.5 μM + T3 10 nM. These concentrations were added to the E3 medium and were replaced daily for 3 or 7 days. Since long-term (7 dpf) treatment with T3 resulted in 100% mortality of the larvae, only Ctrl and IOP conditions were analyzed at this stage. Myelin density was quantified following Black-Gold II staining and the

mRNA expression of the genes involved in the myelination process was measured. As an additional way to visualize the changes in myelin in response to T3 deficiency and restoration, we also subjected *mbp*:*egfp* F0 zebrafish embryos to the above mentioned treatments and analyzed the fluorescent signal. Image analysis and data processing were performed as described above.

### Statistical analysis

Statistical tests were carried out using GraphPad Prism 7 using appropriate tests as specified in each figure legend. The difference was considered statistically significant at $p < 0.05$.

## Results

### Optimization of IOP treatment duration and concentration

We first quantified expression of myelin-related genes (*olig2*, *sox10*, *mbpa*, *mpz*) at various stages of development of the complete zebrafish embryo (0 hpf) and 3, 4, 5 and 7 dpf larvae (**S1 Fig**). We found expression of all but *mpz* at 0 hpf, suggesting that these early transcripts are of maternal origin [23]. For all analyzed genes, larval expression was stable after 3 dpf. Given that myelination processes have commenced by 3 dpf and continue during larval development, we chose 3 and 7 dpf as stages for further detailed analysis.

We next treated zebrafish larvae up to 7 dpf with 0.5, 1, 1.5 and 2 μM IOP to determine the minimum concentration of IOP that would change the expression of these genes. Since all doses clearly had an impact on gene expression (**S1B Fig**) we opted for the lowest dose of 0.5 μM IOP for all further experiments to minimize possible toxic effects.

### Effects of IOP treatment on thyroid status

To corroborate that the IOP treatment was inducing T3 deficiency, we first used Alcian staining, confirming a malformation in Meckel´s cartilage, a characteristic of a hypothyroid status in mammals and fish [42,43], in 7 dpf IOP-treated animals (**Fig 1A**). We then analyzed the expression of genes that are involved in maintaining the thyroidal status [deiodinase type 2 (*dio2*) and deiodinase type 3 (*dio3*)] as well as genes that are known to be TH-dependent [growth hormone (*gh*), thyrotropin (*tshb*) and transthyretin (*ttr*)]. As expected, IOP treatment induced an up- (*dio2*) and down- (*dio3*) regulation of the T3-activating and inactivating pathways, respectively (**Fig 1B**). *gh* and *ttr* expression did not respond to the IOP treatment while that of *tshb* was significantly increased. To corroborate that the selected genes were indeed regulated by THs, we treated 3 and 7 dpf larvae with T3 (10 nM) and analyzed their expression levels. As previously mentioned, the long-term (7 dpf) treatment with T3 resulted in 100% mortality of the larvae. However, treatment in 3 dpf larvae induced *dio3* and *tshb* gene expression changes in the opposite way as compared with IOP-treated larvae. Moreover, *gh* and *ttr* expression increased when exposed to a surplus of T3, demonstrating that TH availability does influence the regulation of these genes (**Fig 1B**). Given the cartilage malformation and discrete responses observed following treatment with 0.5 μM IOP, our results point to a mild hypothyroidism/T3 deficiency.

### Quantification of myelin with Black-Gold II staining

By analyzing specific regions in the forebrain [dorsal telencephalon (D), dorsal thalamus (DT)], midbrain [tectum opticum (TO)], hindbrain [cerebellar plate (CeP) and spinal cord tract (SCT)] in 3 dpf (Ctrl, IOP, T3 and IOP + T3) (**Fig 2**) and 7 dpf (Ctrl *vs* IOP) zebrafish larvae (**Fig 3**) with whole-mount Black-Gold II staining we identified myelinated axons in the form of longitudinal filaments in violet color. We measured the density of the myelin sheaths

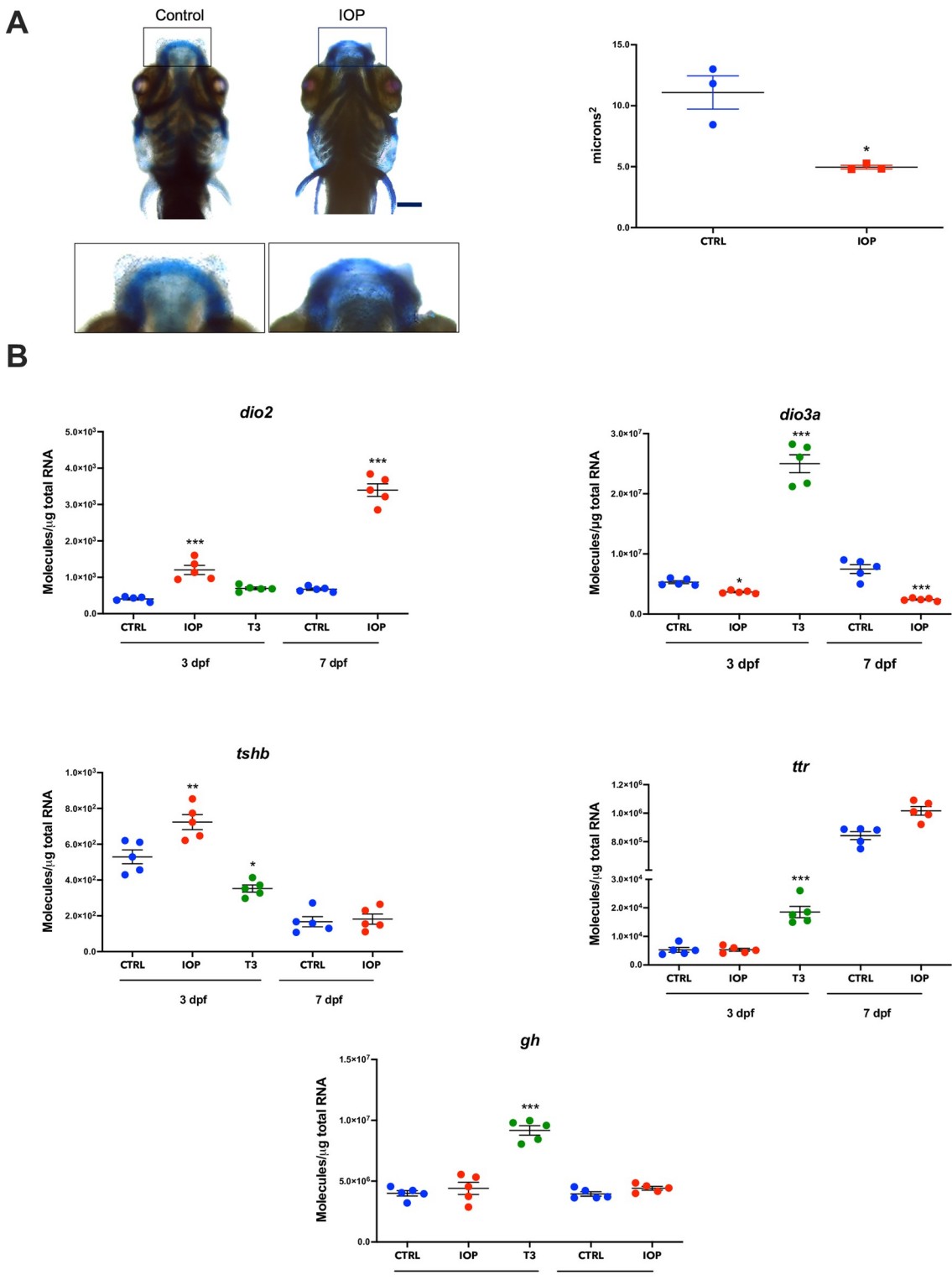

**Fig 1. Effects of IOP treatment on thyroid status.** (**A**) Effect of IOP on Meckel´s cartilage morphology. Left: Representative image of Alcian blue staining of 7 dpf control zebrafish (left) and IOP-treated (right) larvae. Scale bar 100 μm. Right: Graph depicting the quantification of Meckel's cartilage (n = 3 larvae per group). The difference is best visible in the top middle where the stained area is thinner and more faint in the IOP animal. Scale bar 100 μm. Statistical analysis was performed with a Student's t test, *p<0.01 (**B**) Changes in expression of TH-regulated genes in zebrafish larvae exposed to IOP. mRNA quantification of *dio2*, *dio3*, *tshb*, *ttr* and *gh*

in controls and larvae exposed to IOP 0.5 µM or T3 10 nM for 3 dpf and to IOP 0.5 µM for 7 dpf. Data are represented as individual values in vertical graphs (n = 5 pools from independent experiments of 50–60 larvae per pool and condition). Graphs are showing individual values and mean ± SEM. Statistical analysis was performed with one-way ANOVA coupled with Tukey's multiple comparison test with respect to the corresponding control groups. Significant differences are indicated as $^*p < 0.05$, $^{**}p < 0.01$, and $^{***}p < 0.001$.

and detected a strong reduction in the larvae treated with IOP at 3 and 7 dpf (**Figs 2** and **3**). The myelinated area (MA) quantified in pixels in [D + DT + TeO], CeP and SCT (**Figs 2** and **3**) was significantly reduced in IOP-treated larvae, both at 3dpf and 7dpf. Treatment with T3 alone did not change myelin density compared to control, but addition of T3 to the IOP treatment partly rescued the decrease observed with IOP alone.

## Expression of myelin-related genes

As a mean to verify that the observed effects upon myelination were indeed due to the lack of T3, we performed a rescue experiment including the following experimental groups: Ctrl, IOP (0.5 µM), T3 (10 nM) and IOP (0.5 µM) + T3 (10 nM) and analyzed the expression of genes involved in myelination in 3dpf larvae. As mentioned earlier, 7 dpf T3-treated larvae did not survive. IOP treatment resulted in a decreased expression of *olig2*, *mpz* and *plp1b* and T3 treatment up regulated their expression at 3 dpf (**Fig 4**). *sox10* expression increased at 3 dpf in IOP-

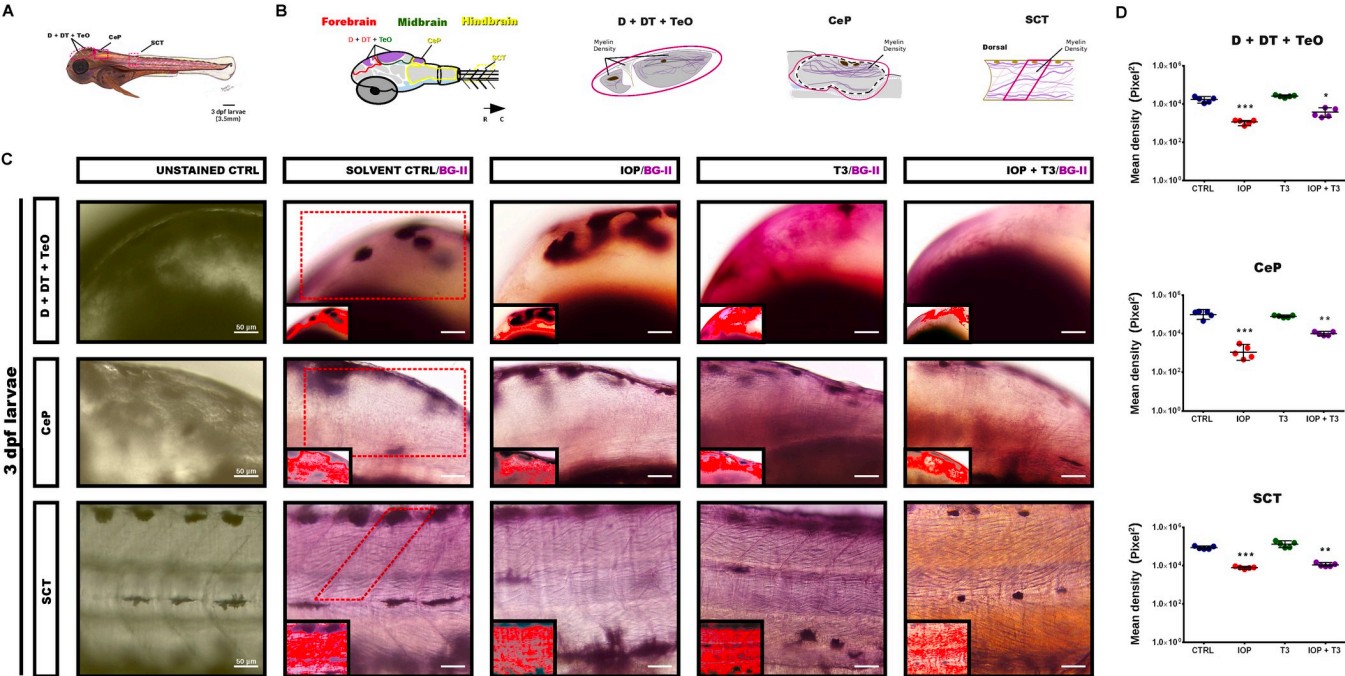

**Fig 2. Whole mount Black-Gold II staining of zebrafish larvae at 3 dpf. (A)** Side view painting of 3 dpf zebrafish larva showing D + DT + TeO; CeP and SCT location; **(B)** Schematic drawing of the brain region (forebrain, midbrain and hindbrain) with identification of myelinated longitudinal axons in a violet color. **(C)** Representative photomicrograph of zebrafish larvae with lateral view and R-C orientation, whole-mount Black-Gold II staining in D + DT + TeO; CeP and SCT at 3 dpf (Unstained CTRL, Solvent Ctrl/BG-II, IOP/BG-II, T3/BG-II and IOP + T3/BGII). In each corner, the amplification of sagittal sections identifying the selected area for analysis and the positive myelin pixels, recognized by the IPP6 program; scale bar: 500 µm and 50 µm. **(D)** Quantification of myelin density expressed as myelinated area (MA) in pixels$^2$ in D + DT + TeO; CeP and SCT in CTRL, IOP, T3 and IOP + T3 larvae (mean ± SEM with n = 5/condition). Statistical analysis was performed with one-way ANOVA coupled with Tukey's multiple comparison test with respect to the control group. $^*p < 0.05$, $^{**}p < 0.01$, $^{***}p < 0.001$.

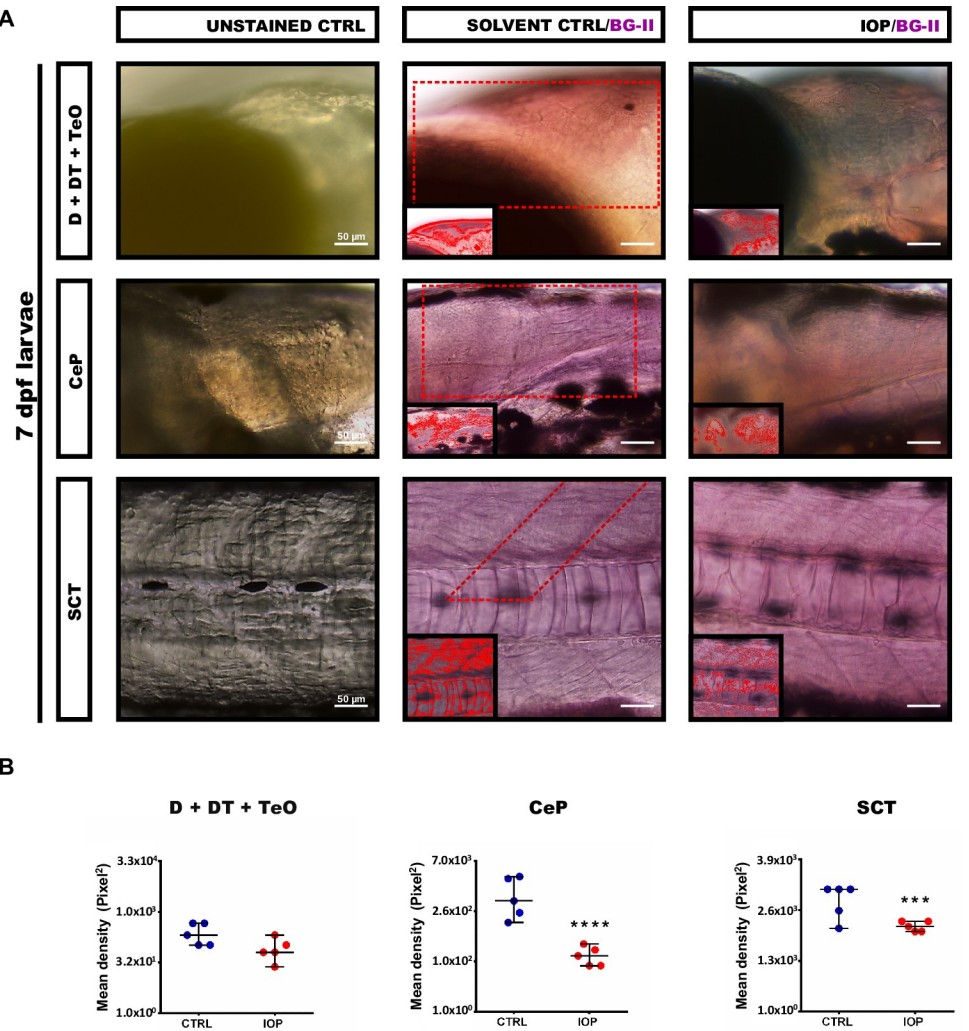

**Fig 3. Whole-mount Black-Gold II staining of zebrafish larvae at 7 dpf. (A)** Photomicrograph with lateral view and R-C orientation, whole-mount Black-Gold II staining in D + DT + TeO; CeP and SCT at 7 dpf (Unstained CTRL, Solvent CTRL/BG-II and IOP/BG-II), scale bar: 500 µm and 50 µm. In each corner, the amplification of sagittal sections identifying the selected area for analysis and the positive myelin pixels, recognized by the IPP6 program; **(B)** Graph showing the quantification of myelin density expressed as myelinated area (MA) in pixels$^2$ (individual values and mean ± SEM with n = 5/condition). Statistical analysis was performed with Student's t test. ***p<0.001, ****p <0.0001.

treated larvae and decreased with T3, while expression of *mbpa* only decreased after 7 days of IOP treatment but was enhanced by T3 in 3 dpf larvae (**Fig 4**).

## Visualization of myelin in *mbp:egfp* zebrafish larvae

As an additional way to visualize the changes in myelin in response to T3 deficiency and restoration we also performed experiments on *mbp:egfp* transgenic animals. We first analyzed *mbp: egfp* signal (voxels) in D + DT + TeO and CeP of 3 dpf zebrafish larvae exposed to increasing concentrations of IOP (0.5, 2 and 5 µM). As the effect was already clear at IOP 0.5 µM (**S2 Fig**), we continued with this concentration for a rescue experiment and analyzed the *mbp:egfp* signal in 3 dpf zebrafish larvae exposed to T3 (10 nM) without or with IOP (0.5 µM). As shown in **Fig 5D** and 5E, the fluorescence emitted by the myelinated regions again decreased

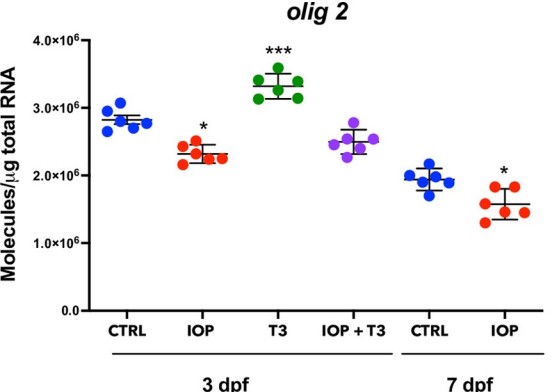

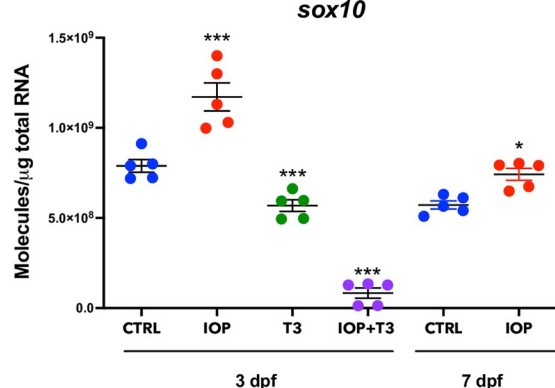

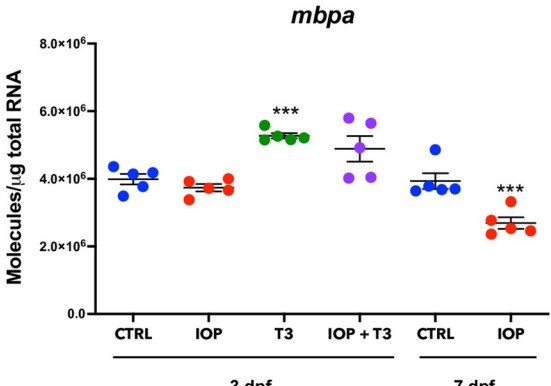

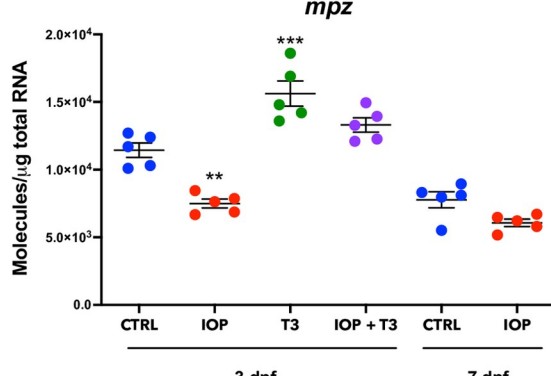

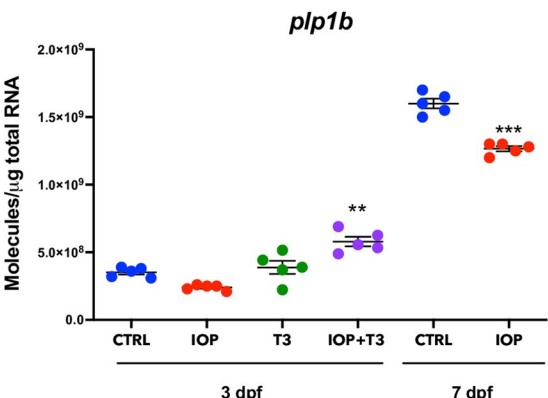

**Fig 4. Changes in expression of myelin-related genes in zebrafish larvae exposed to IOP.** mRNA quantification of *olig2*, *sox10*, *mbpa*, *mpz* and *plp1b* in controls and larvae exposed to IOP 0.5 μM, T3 10 nM or IOP 0.5 μM + T3 10 nM for 3 dpf and to IOP 0.5 μM for 7 dpf. Data are represented as individual values in vertical graphs (n = 5 pools from independent experiments of 50–60 larvae per pool and condition). Graphs are showing individual values and mean ± SEM. Statistical analysis was performed with one-way ANOVA coupled with Tukey's multiple comparison test with respect to the corresponding control groups. Significant differences are indicated as *p $< 0.05$, **p $< 0.01$, and ***p $< 0.001$.

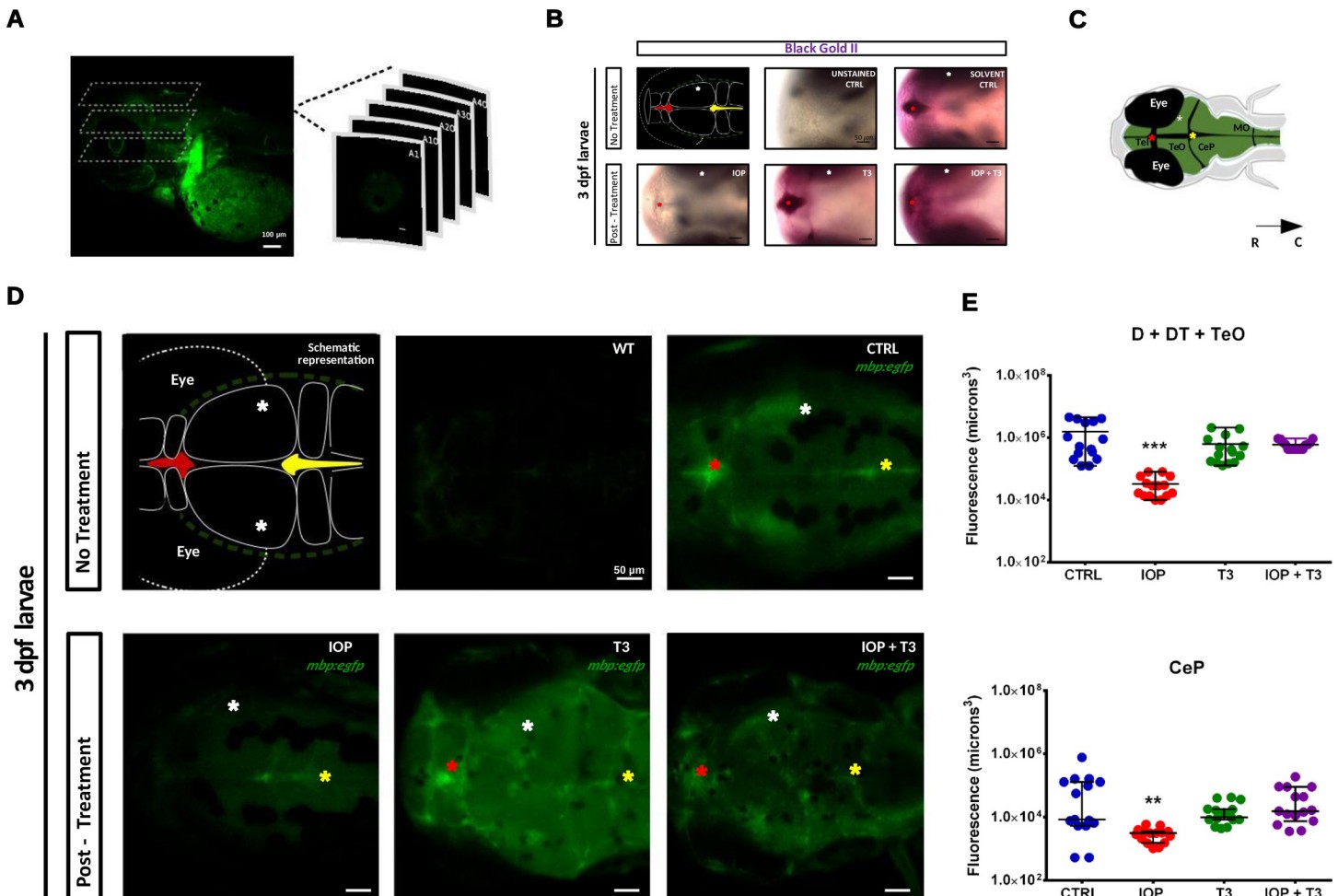

**Fig 5. IOP and T3 rescue treatment in *mbp:egfp* transgenic zebrafish larvae at 3 dpf. (A)** Lateral scan of the head using confocal microscopy and generating multiple 3D optical sections; scale bar: 100 μm. **(B)** Representative photomicrograph of zebrafish larvae with dorsal view and R-C orientation, whole-mount Black-Gold II staining in D, DT at and at 3 dpf (wild-type, CTRL, IOP, T3 and IOP + T3). scale bar: 50 μm. **(C)** Schematic drawing with dorsal view of the head; the asterisks represent signal in the habenula and between the telencephalon (Tel) and tectum opticum (TeO) (red); signal within the TeO (white) and signal in the dorsal Tectum and between TeO and cerebellar plate (CeP) (yellow). The black arrow represents the orientation of the larvae, rostral-caudal (R-C). **(D)** The images are the representation of the maximum intensity projection of the entire set of Z-stacks: uninjected CTRL (WT), CTRL, larvae treated with IOP 0.5 μM, T3 10 nM or IOP 0.5 μM + T3 10 nM; scale bar: 50 μm. **(E)** Graphs depicting the volume of the myelinated area in voxels, Data are shown as individual values and mean ± SEM (approximately n = 15 larvae/group). Statistical analysis was performed with one-way ANOVA coupled with Tukey's multiple comparison test with respect to the control group. **p< 0.01 and ***p< 0.001.

when the larvae were treated with IOP. However, T3 supplementation to IOP larvae increased the fluorescent signal which was restored to levels similar to those of the control group. **S1 Video** exposes the fluorescence inside the brain of zebrafish larvae with a 360˚ rotation.

## OPC development

In order to analyze if T3 deficiency was delaying OPC maturation, we immunostained 3 and 7 dpf control and IOP-treated larvae with the OPC marker NG2. As shown in **Fig 6**, NG2 label is stronger in 3 dpf control larvae, as compared to 7 dpf controls, suggesting a gradual OPC to OL differentiation. IOP treatment reduced the amount of NG2 stain in 3 dpf larvae, as compared to controls, while 7 dpf controls and IOP-treated larvae were no longer significantly different. To further analyze a possible myelination delay *in vivo*, myelin fluorescent signal was followed in living larvae at the onset of myelination (2.5 dpf) and once myelin formation is

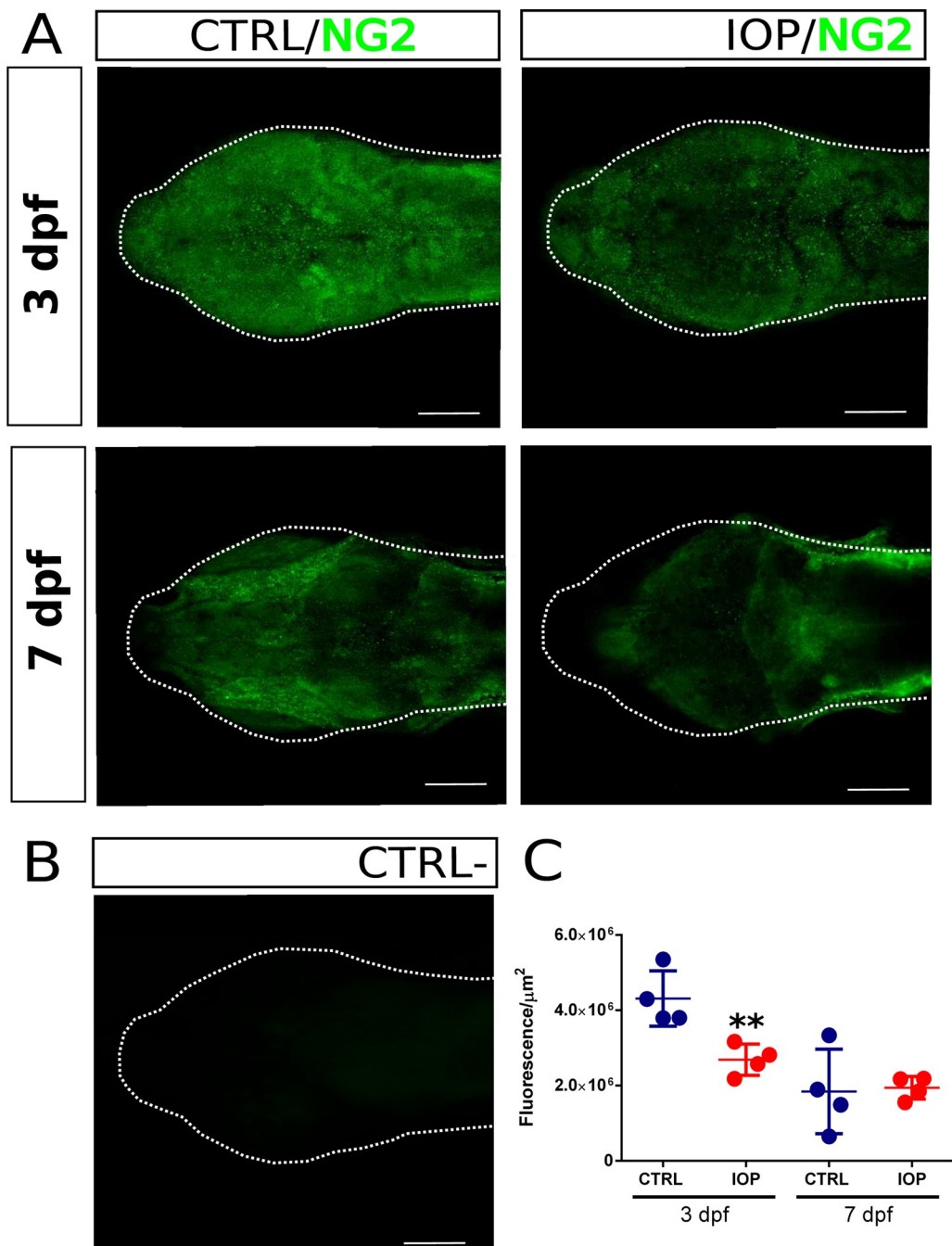

**Fig 6. OPC immunostaining.** Dorsal view of the head of zebrafish larva at 3 and 7 dpf. (**A**) Pictures show whole mount NG2 immunoreactivity in control (upper panel) and IOP-treated larvae (lower panel). The latter show significantly decreased immunoreactivity (**B**) Negative control (without primary antibody) (**C**) Quantification of total fluorescence intensity of NG2 immunoreactivity. Individual values and the mean ± SEM are shown (n = 5 larvae/group). Statistical analysis was performed using Student's t test. Significant differences are indicated as **p< 0.01. Scale bar 100 μm.

ongoing (3 dpf) using *mbp*:*egfp* transgenic larvae with or without IOP treatment. As shown in **S3 Fig**, no myelin was detected in 2.5 dpf controls, while it was clearly visible at 3 dpf. In contrast, 2.5 and 3 dpf IOP-treated larvae did not show a myelin signal.

## Discussion

Thyroid hormones are key modulators of myelination in vertebrates. Hyperthyroidism during the postnatal period results in accelerated myelination, while delayed myelination and poor deposition of myelin is observed in individuals with congenital hypothyroidism [4,10]. In the present study we implemented a protocol that mimics this condition by reducing the conversion of T4, initially of maternal origin, to bioactive T3 and thus limiting its availability during the early stages of neurodevelopment. In the present study we induced a moderate T3 deficiency which allowed to investigate the impact of mildly reduced T3 signaling without too strongly disturbing overall development of the larvae.

When analyzing myelin content in the CNS using Black-Gold II staining, we observed a clear myelin reduction as early as 3 dpf, which became more evident after 7 dpf. A delay in development secondary to IOP treatment could explain myelin reduction; however, we did not observe differences in the length of 3 dpf control and IOP-treated larvae (3.206 ± 0.017 mm$^2$), suggesting that at that stage there was no overall developmental delay and hypomyelination was a specific response to lower T3 bioavailability. Of note is the fact that a decrease in total length was observed in 7 dpf IOP-treated larvae (3.933 ± 0.015 mm$^2$) as compared to controls (3.813 ± 0.027 mm$^2$). Thus, as expected, a longer exposure to reduced T3 impairs both, growth, and myelination. The reduction in the amount of myelin in IOP-treated zebrafish larvae is in line with observations in developing hypothyroid rats, where a clearly lower number of myelinated axons in the anterior commissure and the corpus callosum was described [44]. Another mammalian study reported that hypothyroid neonatal rats had the same number of myelinated axons as control animals but revealed a decrease in the area of these axons [45]. Although more detailed analysis, for instance by electron microscopy, would be needed to draw firm conclusions, data from the present study coincide with studies in which myelin deposition is affected during prenatal and neonatal T3 deficiency [4]. The observed reduction in CNS myelin content did not seem to greatly affect movement, since no differences between control and IOP-treated 7dpf larvae zebrafish were observed, at least not in natural swimming behavior (data not shown).

In the present study the hypomyelination shown by Black Gold II staining was additionally confirmed using *mbp*:*egfp* transgenic animals. We observed a clear reduction in fluorescence in the brain of 3 dpf larvae when T3 availability was low. In concert with the reduction in fluorescence, changes in myelin-related gene expression were also observed. Keeping in mind that analyses were performed on whole larvae RNA and therefore only global net effects could be detected, the results point to a clear effect of T3 deficiency on myelin-related gene expression.

It is known that the onset of myelination starts with OPC formation, which depends at least in part on the expression of transcription factors *olig1/2* and *sox10*, and is followed by mature myelin sheath formation, for which genes involved in myelin composition are expressed (*mbp*, *mpz* and *plp1*) [18]. Additionally, *sox10* is required for the survival of myelinating OLs [16,20]. THs promote OPC differentiation, migration and maturation by increasing expression of Olig2 and Sox10 genes in mammals [46]. In the zebrafish, THs up-regulate the expression of genes encoding structural myelin proteins (*mbp* and *mpz*) and some of the myelination-related genes (*olig2*, *mbp* and *mpz*) were recently shown to contain TH-responsive elements in their promoter regions, as previously shown in mammals [24]. In the present study, 3 dpf IOP-treated larvae showed decreased expression of *olig2*, *mpz* and *plp1b*, genes that are constitutively expressed in CNS mature OL, pointing to a delay in myelination possibly due to a reduced number of mature OL, secondary to TH deficiency, as previously reported in both zebrafish [24,47] and mammals [48,49]. Furthermore, we also observed a peak of expression of

*sox10* at 3 dpf in IOP-treated larvae, suggesting a possible compensatory mechanism to maintain the reduced number of committed myelinating OLs following T3 deficiency.

The hypomyelination observed in IOP-treated *mbp:egfp* transgenic animals was restored after T3 supplementation. These results agree with our previous observations in methimazole-treated juvenile tilapia where the thickness of myelinated axons in the cerebellar granular layer was reduced and could be restored by T3 treatment [50]. T3 rescue also reversed the changes in myelin-related gene expression induced by IOP exposure, evidencing that THs are involved in the molecular pathways regulating myelin formation. The significant increase in *olig2*, *mbpa* and *mpz* and the reduction in *sox10* expression might indicate that the rescue T3 concentration we used was rather high. Given that myelination is established by 3 dpf, an excess of T3 could have a deleterious effect upon myelin-related gene expression. These observations provide further evidence of the importance of TH action during specific time windows in development, and prompt for additional studies to find out whether early exposure to excess T3 only accelerates or possibly disarrays myelin formation.

So far, the specific molecular mechanisms involved in TH-mediated myelination are far from clear. However, and even when THs regulate myelin structural proteins, their effects seem to initially target the differentiation and fate of neural precursor cells destined for myelin formation. Our observations in live *mbp:egfp* larvae showing that T3 deficiency delayed the onset of myelination suggest that OPC number was reduced and/or that myelin synthesis was altered in OLs. The fact that the immunoreactivity for the OPC marker NG2 was also reduced in 3 dpf IOP-treated larvae suggests T3 deficiency could arrest or delay neural progenitor to OPC transition and/or inhibited OPC radial proliferation and migration.

In conclusion, TH deficiency decreased the amount of myelin in different areas of the CNS and affected the expression of myelin-related genes. Since myelination/demyelination/remyelination are dynamic TH-dependent processes, more specific studies are required to better understand the factors involved in OL differentiation and maturation. Established developmental models such as zebrafish are crucial for future avenues of research to address these questions and related demyelination pathologies.

## Supporting information

**S1 Fig. Expression of *olig2, sox10, mbpa* and *mpz* at the onset and establishment of myelin formation during zebrafish development. A.** mRNA quantification at 0 hpf, 3dpf, 4dpf, 5dpf and 7dpf. Results are represented on a logarithmic scale (mean ± SEM; n = 3 pools of 50–60 larvae per stage). **B.** 7 dpf zebrafish larvae were exposed to 0.5 µM, 1 µM, 1.5 µM and 2 µM IOP and mRNA expression of *olig2, sox10, mbpa and mpz* genes was quantified. Statistical analysis was performed with one-way ANOVA coupled with Tukey's multiple comparison test with respect to the control groups. Significant differences are indicated as $^*$p $<$0.05, $^{**}$p $<$0.01, $^{***}$p $<$0.001 and $^{****}$p$<$0.0001.
(TIF)

**S2 Fig. *mbp:egfp* expression in transgenic zebrafish larvae exposed to IOP. (A)** Schematic drawing with dorsal view of the head. The dashed lines mark the brain of the larva (green) and the eyes (white). Red and yellow asterisks refer to the regions that make the division between Tel: Telencephalon, TeO: Tectum opticum, CeP: Cerebellar plate and MO: Medulla oblongata. Confocal images of the larval brain without treatment and with IOP 0.5, 2 and 5 µM treatment are shown. The images are the representation of the maximum intensity projection of the entire set of Z-stacks. **(B)** The graphs show the volume of the myelinated area in voxels. Data are shown as individual values and mean ± SEM (approximately n = 13 larvae/group). Statistical analysis between medians was performed using one-way ANOVA coupled with Tukey's

multiple comparison test. Significant differences (CTRL *vs* Treatment) are indicated as $^*$p< 0.05, $^{***}$p< 0.001 and $^{****}$p< 0.0001.
(TIF)

**S3 Fig. *In vivo* confocal imaging.** Control and IOP 0.5 μM-treated *mbp*:*egfp* transgenic zebrafish were photographed at 2.5 dpf and 3 dpf. At each time point, larvae were anesthetized with Tricaine mesylate (MS-222) 0.2 mg/ml for 45 seconds and fixed and orientated in agarose 1.5%. Confocal images were captured as previously indicated. After caption, animals were returned in a new chamber to their respective treatment. Given that capture was performed in live larvae, the orientation of the head was not always optimal.
(TIF)

**S1 Table. Real time PCR oligonucleotide sequences used to amplify the different genes.**
(PDF)

**S1 Video. 3D reconstruction with Amira software of *mbp*:*egfp* signal in 3dpf zebrafish larval brain.** Dorsal view of the head and brain of zebrafish larvae during the development of brain myelination. The *mbp*:*egfp* signal (represented in yellow) is located between the telencephalon and the TeO as well as between the TeO and the CeP. *Upper images*: Compilation of multiple data from 3D optical sections with signal *mbp*:*egfp* of control and IOP 0.5 μM treated zebrafish larvae; scale bar: 50 μm. *Lower images*: Zebrafish larvae with *mbp*:*egfp* signal, treated with T3 10 nM, and IOP 0.5 μM + T3 10 nM.
(MP4)

## Acknowledgments

The authors would like to thank Dr K. Kawakami for providing the pCS-zT2TP plasmid and Dr. Hae-Chul Park for donating the Tg (*mbp*:*egfp*) plasmid; Dr. Yasmín Guadalupe Hernández Linares, Dr. Edith Espino, Dr. Ericka A. de los Ríos, Ing. Elsa Nydia Hernandez and Lic. Fernando López-Barrera for technical assistance, as well as M. Sc. Gema Martínez Cabrera and Ht. Alfredo Amadeo Díaz Estrada for assisting in the implementation of the histological procedures. The present work received support from Alejandro De León and Carlos Flores from the "Laboratorio Nacional de Visualización Científica Avanzada" (LAVIS) at UNAM-Juriquilla.

## Author Contributions

**Conceptualization:** Aurea Orozco.

**Data curation:** Patricia Villalobos.

**Formal analysis:** Patricia Villalobos.

**Funding acquisition:** Aurea Orozco.

**Investigation:** Brenda Minerva Farías-Serratos, Iván Lazcano, Aurea Orozco.

**Methodology:** Brenda Minerva Farías-Serratos, Iván Lazcano, Patricia Villalobos.

**Project administration:** Aurea Orozco.

**Resources:** Aurea Orozco.

**Supervision:** Veerle M. Darras, Aurea Orozco.

**Validation:** Brenda Minerva Farías-Serratos, Iván Lazcano, Patricia Villalobos, Veerle M. Darras, Aurea Orozco.

**Visualization:** Brenda Minerva Farías-Serratos, Iván Lazcano, Veerle M. Darras.

**Writing – original draft:** Brenda Minerva Farías-Serratos, Aurea Orozco.

**Writing – review & editing:** Iván Lazcano, Veerle M. Darras, Aurea Orozco.

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
