## [Decision Letter · Decision Letter 0]

3 Feb 2021

PONE-D-20-33696

Thyroid hormone deficiency during zebrafish development impairs central nervous system myelination

PLOS ONE

Dear Dr. Orozco,

Thank you for submitting your manuscript to PLOS ONE. After careful consideration, we feel that it has merit but does not fully meet PLOS ONE’s publication criteria as it currently stands. Therefore, we invite you to submit a revised version of the manuscript that addresses the points raised during the review process.

Your manuscript has been reviewed by two referees who are recognized experts in this field. Both reviewers have raised several issues that you will have to consider carefully in order to revise your manuscript accordingly.

We look forward to receiving your revised manuscript.

Kind regards,

Hubert Vaudry

Academic Editor

PLOS ONE

Journal Requirements:

Reviewers' comments:

Reviewer's Responses to Questions

**Comments to the Author**

1. Is the manuscript technically sound, and do the data support the conclusions?

Reviewer #1: Partly

Reviewer #2: Yes

2. Has the statistical analysis been performed appropriately and rigorously? 

Reviewer #1: I Don't Know

Reviewer #2: No

3. Have the authors made all data underlying the findings in their manuscript fully available?

Reviewer #1: Yes

Reviewer #2: Yes

4. Is the manuscript presented in an intelligible fashion and written in standard English?

Reviewer #1: Yes

Reviewer #2: Yes

5. Review Comments to the Author

Reviewer #1: In “Thyroid hormone deficiency during zebrafish development impairs central nervous system myelination,” Farías-Serratos et al. test the role of thyroid signaling in oligodendrocyte myelination. This question is of clinical impact, as human hypo/hyperthyroidism is linked to changes in myelination state. The authors use pharmacology to manipulate thyroid signaling and use a combination of qPCR, transgenic, and histological analyses of myelinating glial cell development. While the implications of this study are interesting and the manuscript is well written, I have several major concerns about the interpretation of the results that would necessitate further investigation before publication. Overall, I am concerned with the Black-Gold stains for assessing myelin in the larvae, as they have so much background that it is difficult to interpret what is the true signal. Below are suggestions for alternatives to improve my confidence in the results:

MAJOR:

1. Assessing oligodendrocyte development by qPCR is difficult to interpret because a number of the markers assessed are expressed in large populations of other cell types: olig2 in motor neurons, sox10 in all neural crest, mbpa in Schwann cells. The authors should take advantage of preexisting transgenic lines or published in situ constructs for these markers, which would be more suitable to address a developmental delay in oligodendrocytes specifically.

2. While the difference in Black-Gold II stain in Figure 2 is obvious, it is impossible to see the difference in the spinal cord (Figure 3) due to the extensive background staining. The authors should perform cross sections of the stained tissue and quantify the signal specifically within the spinal cord.

3. The Black-Gold stain in Figure 4 has so much background that I do not think it actually reflects myelination. At 3 dpf, there is very little myelin present in the brain, so I do not feel confident that the authors are actually quantifying myelin. The authors should take advantage of published in situ probes for mbpa, or rely solely on the mbp:egfp transgene.

4. Assessing oligodendrocyte myelination in single oligodendrocytes through injection of mbp:egfp for mosaic labeling would be useful to determine whether there is a developmental delay (delayed transition to making myelin sheaths), or if fewer oligodendrocytes are being made in general.

5. Given the morphological defects in the IOP-treated animals, the authors should use acetylated tubulin stain to assess whether axon number and/or organization is impaired in these animals.

MINOR:

Line 67: Myelin is not present in all vertebrates. The sentence should be clarified to state that myelination is the last major event to occur in all vertebrates for which axons are myelinated.

Lines 77-79: Mbp is also robustly expressed in Schwann cells. The major difference between central and peripheral myelin is the presence of Plp in oligodendrocytes and Mpz in Schwann cells.

Lines 338-339: Olig2 is not specific to oligodendrocyte precursors. It is also a primary marker of motor neurons (PMID: 12167410), which complicates the interpretation of the qPCR results as noted above.

Statistics: Why were medians compared rather than means?

Reviewer #2: The article by Farias-Serratos et al describes the consequences of pharmacological treatment with iopanoic acid (IOP) on the myelination of zebrafish larvae. The authors treated zebrafish larvae with different concentrations of IOP (and chose the smallest one 0.5 µM) and show deregulation of key genes for thyroid hormone signalling (dio2, dio3, thra, thrb) and oligidendrocyte maturation (olig2, mbp, mpz) in whole embryos. The authors show differences in myelin density with Black Gold II labelling in areas of the central nervous system such as the flat cerebellar or spinal cord tract (SCT) at 3dpf or 7 dpf. However, no significant changes are shown in the diameter or perimeter of the myelinated axons in the regions of the dorsal thalmamus or the ceP.

Rescue experiments are being carried out with T3 on previously deregulated genes ( but not on sections) and mbp :eGFP transgenic animals have been generated to make fluorescence quantification in the presence of IOP and with T3 rescue.

The article is clear and well written. Different parts are well balanced. Discussion is mainly based on a comparison of the data provided in the present article with the data published by Zada on mct8 -/- and could be improved.

I think the article needs minor corrections before publications in PLOS ONE.

1) rationale for the parts of the nervous system is not clear.

2) Even if IOP is known to affect deiodinase activity and thyroid hormoen synthesis , an embryonic treatment with IOP in zebrafish is not well linked with hypothyroidism. As hypothyroidism is not fully demonstrated, some conclusions are not supported by the data presented. Would it be possible to better show the hypothyroidism ?. Can T3 and T4 measurements be performed on embryos (whole or tissue) before and after IOP treatment to support the fundamental hypothesis of hypothyroidism? If not, can a T4 antibody be used? Or study of the thyroid gland? pituitary (TSH) or thyroid growth related genes?

3) The article is mostly descriptive and lacks a functional part to make the data fully usable and understandable.

4) Statistics: with n=3 for the gene exppressions for exemple. A comparison of more than two groups by mann whitney is not recommanded (ANOVA should be used but I don’t know if n=3 is enough for statistical power).

Following are some suggestions that can be taken into account to improve the MS according with editor’s recommandation.

Genes are quantified and related to a total amount of RNA. However, the actin B1 gene is in the list of primers used. It is not clear whether this housekeeping gene has been used for relative quantification or not.

The authors assume that the decrease in dio3 is a sign of hypothyroidism but the increased expression at 3dpf of thrb, mpa at 7 days and the decrease in thraa and thrab do not support this hypothesis. It is also debatable whether these measurements are made on whole organisms. Only an overall net effect is therefore observed and correlations with tissue or even cellular events in the nervous system are hazardous, given that opposite cellular regulations can be observed, particularly during development. A discussion on tissue and cellular effects of thyroid hormone would be therefore interesting

it is quite surprising to see that the T3 treatment alone has not been carried out on gene quantification and it is not described whether the 50/60 embryos x3 come from 3 independent experiments or if they are from the same experiment.

There is no functional experience on the consequences of a delay in myelination (movement for example) and rescue that the T3 treatment could bring. Furthermore, cross sections show a lack of effect of IOP treatment on myelinated axons. How then can this apparent contradiction be explained with the drop in myelin density observed with Black gold II marking? What are the effects of T3 and IOP+T3 treatment on these parameters?

Fig 6/ Video Suppl 4 : the term positive control is not appropriate (solvant control). In the video the size is not the same than the others 3D reconstructions.

A scale bar should be present in all sections/pictures/video

6. PLOS authors have the option to publish the peer review history of their article (what does this mean?). If published, this will include your full peer review and any attached files.

Reviewer #1: No

Reviewer #2: No

---

## [Author Response · Author response to Decision Letter 0]

11 May 2021

Answers to Review

We thank the reviewers for their valuable comments and suggestions. In the new version of the manuscript, we have included additional experiments to address some of the concerns raised, as well as rewritten sections of the discussion. The changes are shown in red throughout the revised manuscript.

Reviewer #1:

1. Assessing oligodendrocyte development by qPCR is difficult to interpret because a number of the markers assessed are expressed in large populations of other cell types: olig2 in motor neurons, sox10 in all neural crest, mbpa in Schwann cells. The authors should take advantage of preexisting transgenic lines or published in situ constructs for these markers, which would be more suitable to address a developmental delay in oligodendrocytes specifically.

and

4. Assessing oligodendrocyte myelination in single oligodendrocytes through injection of mbp:egfp for mosaic labeling would be useful to determine whether there is a developmental delay (delayed transition to making myelin sheaths), or if fewer oligodendrocytes are being made in general. 

We agree with the reviewer that qPCR is not the optimal way to follow up on oligodendrocyte development, considering also that the concentration of the target gene is diluted when mRNA is extracted from the whole embryo. In order to gain some insights regarding a possible delay in oligodendrocyte development due to T3 deficiency, we have included two additional experiments: 1) a follow-up on myelin formation using live mbp:egfp larvae at the onset of myelination and once myelin formation is ongoing; 2) immunostaining using an OPC marker to analyze if OPCs showed a developmental delay with T3 deficiency or if fewer OPCs transitioned to myelinating OLs. (lines 318-330 and 398-406).

2. While the difference in Black-Gold II stain in Figure 2 is obvious, it is impossible to see the difference in the spinal cord (Figure 3) due to the extensive background staining. The authors should perform cross sections of the stained tissue and quantify the signal specifically within the spinal cord. 

We thank the reviewer for the comment. We have repeated the experiment in both, whole-mount preparations and cross-sections. The Black-Gold II staining was improved, resulting in images where myelin is more clearly observed; however, when we performed the Black-Gold II staining in cross-section, images were not optimal to accurately measure myelin content because sectioning harms the integrity of the tissue. Since the new whole-mount images depict myelin content quite unequivocally, we decided to just include these in the new version of the manuscript. Also, we have eliminated the Figure 3 from the original version of the manuscript and have included spinal cord images in the new Figures 2 and 3. 

3. The Black-Gold stain in Figure 4 has so much background that I do not think it actually reflects myelination. At 3 dpf, there is very little myelin present in the brain, so I do not feel confident that the authors are actually quantifying myelin. The authors should take advantage of published in situ probes for mbpa, or rely solely on the mbp:egfp transgene. 

The reviewer is right in that it is difficult to observe myelin in the cross-sections included in Figure 3 of the original version of the manuscript although the magnitude of magnification used while performing myelin determinations allowed the measurements reported in the Figure. However, given the comment and considering that no changes were observed in the analyzed parameters (perimeter and diameter), we decided to eliminate these data from the new version of the manuscript.

5. Given the morphological defects in the IOP-treated animals, the authors should use acetylated tubulin stain to assess whether axon number and/or organization is impaired in these animals. 

We repeated all experiments and found that the IOP-treated animals for 7 days did not consistently show morphological defects. However, to attend the reviewer’s concern, we searched for an antibody to label acetylated tubulin. We finally decided against these experiments since tubulin is a very dynamic structure and we could not find a label that would depict an impairment on axonal organization or number.

MINOR:

Line 67: Myelin is not present in all vertebrates. The sentence should be clarified to state that myelination is the last major event to occur in all vertebrates for which axons are myelinated.

Corrected.

Lines 77-79: Mbp is also robustly expressed in Schwann cells. The major difference between central and peripheral myelin is the presence of Plp in oligodendrocytes and Mpz in Schwann cells.

According to several authors, mpb, mpz and plp1 are distributed in both the central and the peripheral nervous systems. We have corrected the text and included new references to support this notion (Brösamle C, Halpern ME. Characterization of myelination in the developing zebrafish. Glia. 2002 Jul;39(1):47-57. doi: 10.1002/glia.10088. PMID: 12112375).

Lines 338-339: Olig2 is not specific to oligodendrocyte precursors. It is also a primary marker of motor neurons (PMID: 12167410), which complicates the interpretation of the qPCR results as noted above.

Corrected.

Statistics: Why were medians compared rather than means?

While the graph type used (Box and Whiskers) for Figures 1, 4-6 and Supplementary Figure 2 depict medians in the middle of the box, the ANOVA analysis coupled with Tukey test always compares the means. We just used the Box and Whiskers graph type to better show the dispersion of the data. Although we could change the graph type if the reviewer insists, the statistical analysis results remain the same. 

Reviewer #2: 

1) Rationale for the parts of the nervous system is not clear.

We selected regions that included the forebrain, midbrain and hindbrain to have a more integrative picture of CNS myelination. We have now specified the location within the nervous system of the anatomical structures that were analyzed in this study (lines 32, 173-174). 

2) Even if IOP is known to affect deiodinase activity and thyroid hormone synthesis , an embryonic treatment with IOP in zebrafish is not well linked with hypothyroidism. As hypothyroidism is not fully demonstrated, some conclusions are not supported by the data presented. Would it be possible to better show the hypothyroidism? Can T3 and T4 measurements be performed on embryos (whole or tissue) before and after IOP treatment to support the fundamental hypothesis of hypothyroidism? If not, can a T4 antibody be used? Or study of the thyroid gland? pituitary (TSH) or thyroid growth related genes? 

In the present study, we aimed to reduce T3 bioavailability, rather than to induce a classical hypothyroid condition. While embryonic T4 is from maternal origin, most of the embryonic T3 is derived from T4 to T3 conversion. Impairing this conversion with IOP is the only available method to reduce T3 signaling at very early stages, before the embryonic thyroid gland is active. While measuring T4 would not offer proof of a T3 deficiency, measuring T3 would; however, very sensitive methods would be needed to indeed show this partial T3 reduction. To attend the reviewer’s concerns, we instead: 1) set up a protocol to stain and analyze larval cartilage, which is known to reflect the thyroidal status in mammals and fish. Indeed, we observed malformations in Meckel´s cartilage, revealed by Alcian blue staining (Figure 1A); 2) we measured additional TH-linked genes like thsb, ttr and gh. For these and all the genes selected to measure the thyroidal status, we designed new oligos improving their qPCR amplification efficiency. All experiments were repeated, increasing the total number of individuals analyzed per experiment. With these new conditions, we observed a clear TH regulatory pattern for both, dio2, dio3 and tshb (Figure 1B), supporting the lower T3 availability in the larvae. We were unable to detect changes in ttr and gh, suggesting that these genes are not as sensitive to T3 fluctuations, further supporting that with IOP treatment we induced a rather mild T3 deficiency.

3) The article is mostly descriptive and lacks a functional part to make the data fully usable and understandable. 

To address this concern, we included two experiments to gain some insights regarding a possible delay in oligodendrocyte development due to T3 deficiency: 1) a follow-up on myelin formation using live mbp:egfp larvae at the onset of myelination and once myelin formation is ongoing; 2) immunostaining using an OPC marker to analyze if OPCs showed a developmental delay with T3 deficiency or if fewer OPCs transitioned to myelinating OLs (lines 318-330 and 398-406).

4) Statistics: with n=3 for the gene exppressions for exemple. A comparison of more than two groups by mann whitney is not recommanded (ANOVA should be used but I don’t know if n=3 is enough for statistical power).

We thank the reviewer for noting the errors. As mentioned previously, we have repeated all statistical analyses with the suitable statistical tests, running ANOVAs in the experiments where more than 2 groups were compared. Also, the n value in each experiment was increased for a more robust analysis.

Genes are quantified and related to a total amount of RNA. However, the actin B1 gene is in the list of primers used. It is not clear whether this housekeeping gene has been used for relative quantification or not.

We have extended the section on the qPCR methodology for clarity in the new version of the manuscript (lines 137-145). We explain how absolute, not relative gene quantification is performed and the use of two reference genes as correction factors.

The authors assume that the decrease in dio3 is a sign of hypothyroidism but the increased expression at 3dpf of thrb, mpa at 7 days and the decrease in thraa and thrab do not support this hypothesis. It is also debatable whether these measurements are made on whole organisms. Only an overall net effect is therefore observed and correlations with tissue or even cellular events in the nervous system are hazardous, given that opposite cellular regulations can be observed, particularly during development. A discussion on tissue and cellular effects of thyroid hormone would be therefore interesting

As mentioned in the response of point 2, we have included other criteria to determine reduced T3 availability signs in the new version of the manuscript. We have also included a sentence (lines 361-364) to include the arguments raised. However, we decided against discussing on tissue and cellular effects of thyroid hormones in the developing nervous system of zebrafish, given that we do not have detailed results in the present study and that available published information is scarce.

It is quite surprising to see that the T3 treatment alone has not been carried out on gene quantification and it is not described whether the 50/60 embryos x3 come from 3 independent experiments or if they are from the same experiment.

In the newly performed experiments, we have included results from gene quantification of T3- (Figure 1) and IOP + T3-treated groups (Figures 1 and 4). We have also clarified that each pool represents an independent experiment (lines 582 and 615).

There is no functional experience on the consequences of a delay in myelination (movement for example) and rescue that the T3 treatment could bring. Furthermore, cross sections show a lack of effect of IOP treatment on myelinated axons. How then can this apparent contradiction be explained with the drop in myelin density observed with Black gold II marking? What are the effects of T3 and IOP+T3 treatment on these parameters?

Indeed, cross sections did not show a decrease in axonal diameter. In order to confirm these results, we repeated the experiment in both, whole-mount preparations and cross-sections. The Black-Gold II staining was improved, resulting in images where myelin is more clearly observed; however, when we performed the Black-Gold II staining in cross-section, images were not optimal to accurately measure myelin content because sectioning harms the integrity of the tissue. Since the new whole-mount images depict myelin content quite unequivocally, we decided to just include these in the new version of the manuscript. For this reason, we have eliminated the Figure 3 of the original version of the manuscript and have included new images in Figures 2 and 3. In them, we observe a clear decrement in myelin content after IOP treatment at 3 and 7 dpf. However, this decrement is not reflected in movement. In our hands, we found no differences between control and IOP-treated 7dpf zebrafish larvae in natural swimming behavior using single visual inspection and through a free software analysis (Kinovea, ToxTrac). The analysis of the effects of T3 and IOP+T3 was not possible because 7 dpf larvae died during the experimental treatment, as mentioned in the new version of the manuscript. We have included this information in lines 353-356.

Fig 6/ Video Suppl 4: the term positive control is not appropriate (solvant control). In the video the size is not the same than the others 3D reconstructions.

A scale bar should be present in all sections/pictures/video

We have corrected the video.

---

## [Decision Letter · Decision Letter 1]

16 Jun 2021

PONE-D-20-33696R1

Thyroid hormone deficiency during zebrafish development impairs central nervous system myelination

PLOS ONE

Dear Dr. Orozco,

Thank you for submitting your manuscript to PLOS ONE. After careful consideration, we feel that it has merit but does not fully meet PLOS ONE’s publication criteria as it currently stands. Therefore, we invite you to submit a revised version of the manuscript that addresses the points raised during the review process. 

Your revised manuscript has been reviewed by the same two reviewers. Both reviewers mentioned that your manuscript has been improved but were not fully satisfied with the revision. You will have to carefully consider their recommendations and re-revise your manuscript accordingly. At this stage, I cannot guarantee that your re-revised manuscript will be found acceptable for publication in Plos One.

We look forward to receiving your revised manuscript.

Kind regards,

Hubert Vaudry

Academic Editor

PLOS ONE

Reviewers' comments:

Reviewer's Responses to Questions

**Comments to the Author**

1. If the authors have adequately addressed your comments raised in a previous round of review and you feel that this manuscript is now acceptable for publication, you may indicate that here to bypass the “Comments to the Author” section, enter your conflict of interest statement in the “Confidential to Editor” section, and submit your "Accept" recommendation.

Reviewer #1: (No Response)

Reviewer #2: (No Response)

2. Is the manuscript technically sound, and do the data support the conclusions?

Reviewer #1: Partly

Reviewer #2: Partly

3. Has the statistical analysis been performed appropriately and rigorously? 

Reviewer #1: Yes

Reviewer #2: Yes

4. Have the authors made all data underlying the findings in their manuscript fully available?

Reviewer #1: No

Reviewer #2: Yes

5. Is the manuscript presented in an intelligible fashion and written in standard English?

Reviewer #1: Yes

Reviewer #2: Yes

6. Review Comments to the Author

Reviewer #1: In this revised version of “Thyroid hormone deficiency during zebrafish development impairs central nervous system myelination,” Farías-Serratos et al. expand their analysis of the role of thyroid signaling in oligodendrocyte myelination. The implications of this study are interesting and the manuscript is well written, and the additional experiments (assessment of OL development) have added support to their initial claims. I have several concerns that need to be addressed prior to publication, mostly related to statistical rigor.

1. For the new Alcian blue stains (Figure 1A-B), the authors should show labeling and quantification of the cartilage malformation. To the general reader, this difference is not immediately apparent.

2. For all graphs, individual data points she be plotted within the bar graph or box/whisker plot to increase transparency of reporting.

3. The control images for CeP and D+DT+TeO in Figure 2 in the resubmission are not nearly as clear as Figure 2 in the original submission. Please provide representative images with less background stain.

MINOR:

1. Line 68: Myelination in mammals starts in the late stages of embryonic development, not during early stages of development.

2. Line 97-100. Need a citation or else need to state that this is the hypothesis.

3. Line 101 typo. Should read “post-fertilization in teleost”

4. Line 104 typo. Should read “later of larval”

5. Lines 126-127 are redundant with lines 121-122.

6. Lines 284/285, it would be helpful to have these structures defined for the first time here rather than in the methods.

7. Figure 5: need to label the transgene used and the developmental stage.

Reviewer #2: The review of the paper by Farias-Serratos et al on the effect of iopanoic acid on myelin was extensively addressed by the authors.

The quantitative PCRs are now clearly shown and appropiately detailed. The effects are unambiguous. The effects on tsh suggest an IOP-induced decrease in thyroid hormone and a T3-induced increase at 3dpf.

Black gold staining of myelin at 3 days and 7 days showed a decrease in myelin observed at 3 days and still visible at 7 days.

The results at 3 days are not always very convincing since only developmental delay could explain the absence of myelin in the fin . I personally find it very difficult to see a difference between BGII stained animals , without the fibers drawn in red in figures 2 and 3.

IHC with NG2 on larvae in dorsal view seem diffuse and the question of specificity arises. The authors also added experiments using the mbp:eGFP transgenic line. It would have been useful to confirm the data in Figure 3 with this model (given the paper by Jung et al 2010 and the specificity observed in the fin but not necessarily in the brain). Moreover, the signal seems very diffuse. Finally the structures, especially the telencephalon are clearly different when comparing IOP, CTRL and T3 and here again a growth delay explaining the absence of myelin cannot be ruled out.

Overall I recognize the efforts that the authors have made and the quality achieved for the transcriptomic data is very satisfactory. The article reads very well and the reviewers' modifications were generally taken into consideration. There remains this point about the brain structures, which are different (and differently myelinated) depending on the stage of development and/or treatments.

line 227 : This is not true that IOP and T3 treatment always induce gene expression going to opposite ways. For example, for dio2, they go the same sense. Actually this statement is only valid for dio3 and tshb expression.

Also in fig 4 it seems quite weard that T3+IOP is not more fluorescent than CTRL given the very high fluorescent signal in TeO.

7. PLOS authors have the option to publish the peer review history of their article (what does this mean?). If published, this will include your full peer review and any attached files.

Reviewer #1: No

Reviewer #2: No

---

## [Author Response · Author response to Decision Letter 1]

9 Jul 2021

We thank the reviewers for their careful revision of our manuscript. As suggested, all Figures were modified to include individual data points and/or to provide with neater and clearer representative images. Also, attending the comments raised by Reviewer #2 regarding the possibility that the absence of myelin could be related to a developmental delay due to the IOP treatment, we measured control and IOP-treated larvae at 3 and 7 dpf. As discussed in the point-by-point rebuttal, we observed that lower T3 bioavailability does impair correct myelination and that a delay on growth is not evident until 7 dpf.

---

## [Decision Letter · Decision Letter 2]

3 Aug 2021

Thyroid hormone deficiency during zebrafish development impairs central nervous system myelination

PONE-D-20-33696R2

Dear Dr. Orozco,

We’re pleased to inform you that your manuscript has been judged scientifically suitable for publication and will be formally accepted for publication once it meets all outstanding technical requirements.

Kind regards,

Hubert Vaudry

Academic Editor

PLOS ONE

Additional Editor Comments (optional):

Reviewers' comments:

Reviewer's Responses to Questions

**Comments to the Author**

1. If the authors have adequately addressed your comments raised in a previous round of review and you feel that this manuscript is now acceptable for publication, you may indicate that here to bypass the “Comments to the Author” section, enter your conflict of interest statement in the “Confidential to Editor” section, and submit your "Accept" recommendation.

Reviewer #1: All comments have been addressed

Reviewer #2: (No Response)

2. Is the manuscript technically sound, and do the data support the conclusions?

Reviewer #1: (No Response)

Reviewer #2: Partly

3. Has the statistical analysis been performed appropriately and rigorously? 

Reviewer #1: (No Response)

Reviewer #2: Yes

4. Have the authors made all data underlying the findings in their manuscript fully available?

Reviewer #1: (No Response)

Reviewer #2: Yes

5. Is the manuscript presented in an intelligible fashion and written in standard English?

Reviewer #1: (No Response)

Reviewer #2: Yes

6. Review Comments to the Author

Reviewer #1: (No Response)

Reviewer #2: I have seen this artilce for the third time. The authors changed the figures, again .

I feel that indivual plotting is a very clear way to present data. I am still not convinced by the BGII staining especially in the forebrain where it shoudl not be seen myelination at 3dpf, according to the literature and to what authors mention in the introduction. I regret the authors did not acceed to my query of using trangenic animals. Even though I think it could have stregnthen the article I do not want to spend more time on this. Therefore I have last recommandations authors may use before publication.

IOP does not only block T4 to T3 conversion (as stated in the abstract) but also T3 inactivation. Even though I understand this is the activity the authors focused on, the IOP blocking IRD activity should be mentioned in the introduction (especially when presenting IOP+T3 results).

Line 276-277 ; treatment in 3 dpf larvae induced dio3 and tshb gene expression

Line 277 changes in the opposite way as compared with IOP-treated larvae. _The way the sentence is written a bit missleading cause tshb is decreased by T3.

Black gold staining :

To state the there is a partial rescue column to column comparison should be done between IOP and IOP+T3 groups.

There are opposite results showed by black gold staining (no T3 effects vs strong effects on qpcr). IOP+ T3 treatment reveal that T3 by itself has no effect in contrast to what could be seen at molecular level. This is partially discussed (maximum reached in CTRL and T3 but T3 accelerates the process), however it seems weard that T3 by itself has no global effect (BGII Staining or GFP)

Line 385-386 : plp1b is not seen as significatly down regulated in figure 4

Line 391-193 : This is an interesting suggestion that sox10 reveals uncomitted myelinated cells cells but how explaining IOP+T3 effect? In my opininon IOP+T3 is a super T3 effect ( no degradation of a part of T3 so more T3 available). However this is contradictory with mbp egFP results where you see a T3 reversal IOP induced effect. This should be discussed ? (following paragraph in the discussion).

Line 407 : If T3 accelerates myelination, does a significant difference be seen in mbp :egpf larvae ? as quantification is done a 3dpf ?

7. PLOS authors have the option to publish the peer review history of their article (what does this mean?). If published, this will include your full peer review and any attached files.

Reviewer #1: No

Reviewer #2: No

---

## [Editor Report · Acceptance letter]

6 Aug 2021

PONE-D-20-33696R2 

Thyroid hormone deficiency during zebrafish development impairs central nervous system myelination 

Dear Dr. Orozco:

I'm pleased to inform you that your manuscript has been deemed suitable for publication in PLOS ONE. Congratulations! Your manuscript is now with our production department. 

Kind regards, 

on behalf of

Dr. Hubert Vaudry 

Academic Editor

PLOS ONE